ecology

multiple hypotheses, simulation models, modelling, inference, scientific method

**Author for correspondence:**
Scott W. Yanco
e-mail: scott.yanco@ucdenver.edu

# A modern method of multiple working hypotheses to improve inference in ecology

Scott W. Yanco[1], Andrew McDevitt[1], Clive N. Trueman[2], Laurel Hartley[1] and Michael B. Wunder[1]

[1]Department of Integrative Biology, University of Colorado Denver, Denver, CO, USA
[2]Ocean and Earth Science, University of Southampton, National Oceanography Centre, Southampton, UK

SWY, 0000-0003-4717-9370; AM, 0000-0002-8446-4931; CNT, 0000-0002-4995-736X; LH, 0000-0002-2001-1108; MBW, 0000-0002-8063-2408

Science provides a method to learn about the relationships between observed patterns and the processes that generate them. However, inference can be confounded when an observed pattern cannot be clearly and wholly attributed to a hypothesized process. Over-reliance on traditional single-hypothesis methods (i.e. null hypothesis significance testing) has resulted in replication crises in several disciplines, and ecology exhibits features common to these fields (e.g. low-power study designs, questionable research practices, etc.). Considering multiple working hypotheses in combination with pre-data collection modelling can be an effective means to mitigate many of these problems. We present a framework for explicitly modelling systems in which relevant processes are commonly omitted, overlooked or not considered and provide a formal workflow for a pre-data collection analysis of multiple candidate hypotheses. We advocate for and suggest ways that pre-data collection modelling can be combined with consideration of multiple working hypotheses to improve the efficiency and accuracy of research in ecology.

## 1. Replication crises and inferential frameworks

The ultimate goal of science is to learn about the relationships between observable patterns in the world around us and the processes that generate those patterns. Most commonly, scientists identify and/or quantify the links between process and pattern by hypothesizing the existence of a particular relationship between the two and using data as evidence for or against that hypothesis. However, inference may be unreliable if the scientist does not consider all potentially relevant processes. For example, inference

is confounded when unconsidered hypotheses produce the same observed pattern as the stated hypothesis. Similarly, inference is muddled when hypotheses overlook additional variance-inflating processes, effectively rendering the link between process and pattern indiscernible. In either case, researchers who do not carefully guard against such pitfalls may make inferences that are either too strong or too weak.

Recently, several scientific disciplines have experienced 'replication crises' (e.g. cancer biology [1] and psychology [2] among others [3]). Many factors have probably contributed to replication crises: publication bias [4,5], hypothesizing after results are known [6], $p$-hacking [5,7] and data fabrication [8] to name a few. In addition to these factors, irreproducibility has also been driven by an over-reliance on null hypothesis significance testing (NHST; [1,9–12]). The limitations, misuse and outright abuse of NHST are myriad and, by now, well known (see, for example, [5,12–15]). NHST produces erroneous inference both because it is frequently misinterpreted by researchers [12,14,16–18] and because it is prone to manipulation [5,14].

One potentially underappreciated limitation of NHST is that it does not produce evidential support for hypotheses, instead providing only weak evidence of incongruence between observed data and a null hypothesis [12]. The ubiquitous $p$-value quantifies only the probability of hypothetical future data resulting in some summary statistic that would be less consistent with summary statistics computed from data generated by the null hypothesis. If that probability is sufficiently low (e.g. $p < 0.05$), the researcher 'rejects' the null hypothesis as having been unlikely to generate the observed data (as in [19]). Often, 'rejection of the null' leads (illogically) to acceptance of whatever was proposed as the alternative to that strawman; the alternative hypothesis is accepted without any positive inferential support [14]. Furthermore, the NHST framework considers only a single hypothesis. Indeed, the complement to the null hypothesis comprises a *set* of alternative hypotheses. In other words, a 'significant' significance test indicates that data like ours are improbable given a single null hypothesis [12,14]—it produces no information about the infinite number of possible alternative hypotheses [20]. Imagine the potential for error when an automatically accepted alternative hypothesis is not uniquely distinguishable from some other hypothesis the researcher never considered.

Here, we describe methods for considering multiple hypotheses by advocating for the implementation of multi-hypothesis modelling *prior* to data collection. Akin to *in silico* experimentation, design phase modelling helps to identify a plausible set of candidate hypotheses and determine which of the set might lead to any of several different observable patterns [21,22]. Below, we detail the nature of problematic sets of hypotheses and draw on the oft-invoked 'method of multiple working hypotheses' [23] as a partial solution. This method has been repeatedly invoked as an important component of good scientific practice (e.g. [24–26]). In this paper, we propose a workflow invoking the method of multiple working hypotheses in the context of pre-data collection modelling. Our workflow applies recommended practices in theoretical modelling to the problem of design phase modelling with particular emphasis on the consideration of multiple hypotheses. The practical recommendations in our approach are intended to facilitate wider adoption of multiple hypothesis methods, guard against inferential errors to which multi-hypothesis methods are still prone and provide a formal framework for such analyses. This combination of multi-hypothesis inference and pre-data collection modelling represents a powerful alternative incarnation of the scientific method geared towards stronger inference that is less susceptible to errors arising from unconsidered processes.

Specifically, we outline five steps for vetting hypotheses. These steps can be repeated iteratively until the proposed mechanisms and observation patterns adequately map to one another (figure 1). The steps are:

1. specify candidate hypotheses;
2. write a model for each hypothesis;
3. generate sampling distributions of simulated data from each hypothesis;
4. quantify the variance within and overlap between sampling distributions; and
5. revise hypotheses as necessary and repeat steps 1–4.

## 2. The effects of unconsidered alternative hypotheses

Scientific inference and, in particular, inference using NHST assumes that processes are uniquely identifiable from the observable patterns they generate (figure 2). That is, they depend on the statistical concept of identifiability. Model parametrizations are identifiable if and only if distinct parametrizations lead to different probability distribution functions [27].

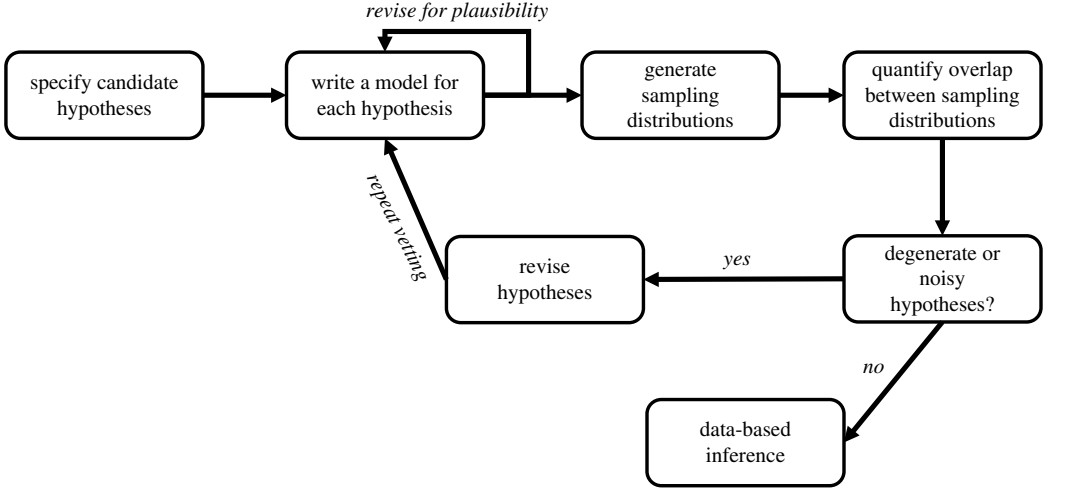

**Figure 1.** Conceptual flow chart of hypothesis vetting process. Researchers first specify a set of candidate hypotheses to consider before writing them as formal models. Formal models are checked for internal coherence and revised, if necessary. Sampling distributions of simulated response variables are generated from each candidate hypothesis which can then be compared to one another for evidence of degeneracy or noisiness. If no such problems exist, the researcher proceeds with data-based inference. Alternatively, the researcher revises the set of candidate hypotheses and begins the hypothesis vetting anew.

Muddled inference (i.e. non-identifiability) manifests in two ways: (i) degenerate relationship: multiple processes produce indistinguishable patterns, or (ii) noisy relationship: processes do not reliably produce a single identifiable pattern (figure 2; [28,29]).

## 2.1. Degenerate relationship

Hypotheses with degenerate relationships between pattern and process are not testable—a fundamental requirement to differentiate hypotheses. In degenerate cases, a single observed pattern could have been produced by more than one process (figure 2; [28,29]). Thus, no single process can be uniquely implicated by the observation. Degeneracy may occur because unconsidered deterministic or stochastic processes modify the resultant pattern. At its heart, this phenomenon arises due to model misspecification wherein two or more models (hypotheses), as specified by the researchers, produce indistinguishable response patterns [29]. In this situation, no observation can serve as evidence of any unique process because multiple processes could have produced the same pattern.

## 2.2. Noisy relationship

Noisy relationships are those wherein a single mechanism produces multiple and varied response patterns potentially due to unrecognized or unconsidered mechanisms (figure 2). Too much variance leads to low predictive power and imprecise estimates of model parameters [27]. Like the degenerate relationship problem, this also results in the same muddled inference. Noisy relationships between patterns and processes commonly arise from observation or measurement errors, or from a mis-specified model.

# 3. The method of multiple working hypotheses revisited again

While inferential failures leading to replication crises have garnered much recent attention [15], they are hardly new. Cohen [30] pointed out flaws in NHST over 25 years ago—and in so doing reminded readers that Bakan [31] made similar arguments over 30 years prior to that. In 1964, Platt described 'strong inference' which grounded much of what Ioannidis [9] demonstrated over 40 years later. In fact, as early as 1890, Thomas Chamberlin described the 'method of multiple working hypotheses' and it has since been repeatedly advocated as a way to mitigate the risk of omitting potentially relevant processes from inference [23–25,32].

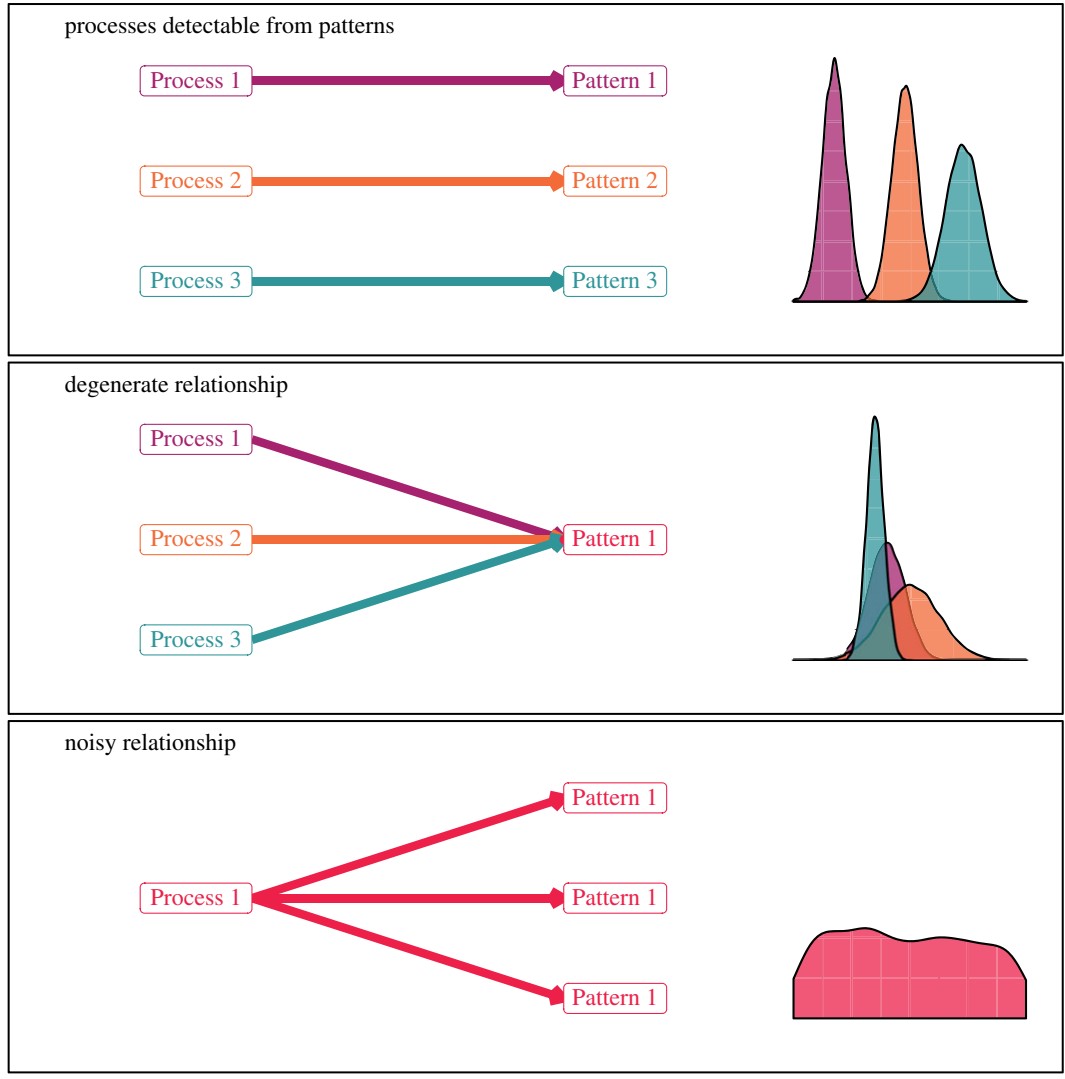

**Figure 2.** Heuristic relationships between processes and observable patterns that drive inferential outcomes. Boxes linked by arrows represent individual hypotheses that can or cannot be parsed based on observed patterns. Density plots show the examples of sampling distributions arising from each hypothesis. Processes detectable from patterns: for a hypothesis to be testable, the response patterns must reliably, quantifiably and uniquely correspond to the hypothesized mechanisms. Note how each process is uniquely linked to a distinct pattern with little or no overlap between sampling distributions. Degenerate relationship: multiple mechanisms degenerating to an indistinguishable response pattern. Each unique process leads to the same observation pattern, and sampling distributions are almost completely overlapping. Noisy relationship: a single mechanism does not reliably produce a concordant response pattern. A single hypothesized process leads to a widely varying response pattern. High variance and/or multi-modal sampling distribution makes estimation difficult or impossible.

Chamberlin [23] advocated that, to avoid foreseeable inferential errors, researchers should explicitly consider *multiple* working hypotheses from the outset. The method is intended to reduce cognitive biases which cause researchers to only collect evidence for favoured hypotheses. Additionally, Chamberlin points out that single-hypothesis frameworks fail to adequately account for complex systems wherein multiple processes may play causal roles—as is common in ecology [23,25]. Using this method, a researcher 'competes' evidence about as many hypotheses as are plausible rather than simply considering the evidence against a strawman hypothesis (as in NHST).

Despite at least 130 years of advocacy for multi-hypothesis approaches, consideration of multiple hypotheses in ecology continues to be rare [33]. For example, Betini *et al.* [26] found that only 21% of a sample of recently published papers in ecology and evolution considered multiple hypotheses. Yet, the systems investigated in these fields are precisely those which stand to benefit from multi-hypothesis approaches (i.e. those involving multiple interacting causal factors; [34]).

Observable patterns arising from myriad interacting variance-generating processes is the norm in ecology. Such complex causal structures are prone to both the degenerate and noisy relationship problems [34]. Consider just a few examples chosen from sub-disciplines within ecology: Boeklen *et al.* [35] identified at least 44, hierarchically organized, factors that influence emergent patterns of tissue stable isotopes used in trophic ecology studies. Several authors have observed sufficient variance in species distributions to produce absurd or impossible model fits [36]. For example, Fourcade *et al.* [37] demonstrated that rasterized paintings projected onto landscapes provided comparable or better fitting models for species distributions than real environmental variables (see also box 1 and electronic supplementary material, 1). Finally, Nathan *et al.* [40] showed that animal movements emerge from an interaction between the organism's motility, capacity to navigate, internal state and external environmental setting—each component of which may themselves entail multiple interacting variables (see electronic supplementary material, 2). These are examples of fields wherein identifying mechanistic drivers via observed patterns is challenging because of the multifaceted nature of the problems at hand—a ubiquitous scenario in ecology. As such, establishing the identifiability of the set of plausible hypotheses should be regarded as the default first step towards reliable inference in ecology.

Compounding the effects of underlying complexity, problematic inferential practices may be common in ecology. For example, Fraser *et al.* [41] found that questionable research practices were widespread—observing rates comparable to fields whose replication crises are well established. In fact, recent, large-scale studies have shown that early, low-power findings in some sub-fields apparently do not replicate (e.g. [42]). In combination, these facts make clear that ecology must improve its inferential toolbox.

Frameworks that support data-based inference between multiple hypotheses are well established (e.g. information theoretic approaches to multi-model inference, [13,33,43–47]), and have even been explicitly linked to Chamberlin's method [25]. Yet, while some sub-disciplines within ecology have seen wider adoption of these tools [25], clearly they remain underused [26,33]. More importantly, *a posteriori* multi-hypothesis methods cannot disentangle hypotheses that are confounded *a priori* (i.e. those that are structurally non-identifiable). As such, employing Chamberlin's method at the earliest stages of research may improve inference. Therefore, pre-data collection modelling is an essential first step in considering multiple hypotheses.

# 4. Pre-data collection modelling enables the method of multiple working hypotheses

Models constructed prior to data collection can provide insights allowing researchers to quantify and, ultimately, to increase clarity and transparency about hypotheses [22,48,49]. These models are essentially *in silico* experiments which simulate response variables using predefined parameters taking on biologically relevant values. Specifying a model forces the researcher to explicitly consider the nature of linkages between the process(es) under investigation and the pattern(s) observed [49]. By using biologically defined parameters, the simulated pattern is clearly understood because the structural components of the model explicitly represent biological links between process and pattern [22]. Comparing simulated responses across multiple alternative hypotheses allows a researcher to quantify the identifiability of each candidate model.

This step, though formally distinct, is analogous to a power analysis wherein researchers use pre-data collection models to ensure that the proposed sample will be sufficient to answer the question at hand. Whereas a power analysis assesses the sufficiency of sample sizes (given some assumed effect size), our framework assesses the identifiability of each hypothesis. Both analyses are ways to ensure, at the outset, that a proposed study is even theoretically capable of producing an answer.

Modelling in this context embraces the method of multiple hypotheses: researchers consider not only a favoured hypothesis but also alternative formulations. This uncovers situations wherein multiple processes might produce observable patterns that are indistinguishable from one another. Of course, engaging in this process does not guarantee that all possible processes will be identified. In fact, there always remains the possibility of a plausible hypothesis a researcher has yet to consider. Betini *et al.* [26] describe typical cognitive barriers that prevent researchers from articulating a complete set of multiple working hypotheses (e.g. lack of creativity, lack of time or incentives to expend the effort, lack of practice with or comfort with brainstorming alternative hypotheses, etc.). The workflow we propose below does not, by itself, overcome those cognitive challenges but we believe it provides a facilitating framework.

# 5. A workflow for vetting multiple working hypotheses

We term the process of modelling multiple hypotheses in the design phase 'hypothesis vetting'. The outcome of hypothesis vetting is to determine whether each candidate hypothesis is uniquely identifiable. Hypothesis vetting is carried out by formally defining variance-generating process(es) and the pattern(s) they produce for each competing hypothesis. In simple systems, this can be accomplished using an analytically tractable equation or set of equations. In more complex systems, this process may require numerical approaches or researchers might instead employ algorithmic simulation models (particularly stochastic models and/or agent-based models).

Each hypothesis is modelled as a unique combination of processes (i.e. variables) and/or a unique combination of linkages between processes and patterns (specific parametrizations). Subsequently, the comparisons of simulation outputs quantify the degree to which patterns can uniquely identify hypothesized process(es). Importantly, this approach provides inference only about the ability to differentiate between simulation models (as in [22]), and not about the validity of any specific model itself.

Below we describe each step in the workflow; box 1 contains a stylized example of implementing this workflow (R code, using the *checkyourself* package, for this example and another is contained in electronic supplementary material, 1 and 2). The provided example is drawn from spatial ecology, but the workflow could (and should) be extended/applied across the diverse sub-disciplines in ecology. For a non-spatial example, we also refer readers to Vagle and McCain [50] who demonstrated *a priori* degeneracy between competing hypotheses about the mechanisms underlying primary productivity–diversity relationships.

## 5.1. Step 1: specify candidate hypotheses

To vet hypotheses, a researcher first conceives of the set of candidate hypotheses. Importantly, the researcher ought not only specify their favoured hypotheses but should specify as many additional plausible hypotheses as possible [23]. The set of hypotheses should consider both alternative combinations of processes and alternative linkages between these processes and resultant patterns (i.e. parametrizations). Further, observation error and study design elements can (and often should) be included as components of hypotheses since they influence the pattern that is ultimately observed.

The complexity of ecological systems has led to some criticism of multi-hypothesis approaches in the field. For example, Simberloff [51] criticized Platt's method [24] on the basis that 'strong inference' is incompatible with ecological processes typically involving multiple, non-mutually exclusive additive or interacting causative factors. This argument rests on the notion that multi-causal systems cannot be subjected to Popperian falsification (as in [52]) because no sufficient model can be written for the hypothesis; as the falsification process proceeds, the ecologist is ultimately left with a set of inseparable, and therefore, unfalsifiable causal factors which all have relevance [51]. Several authors correctly point out that alternative epistemological frameworks accommodate this complexity by estimating probabilistic support *for* hypotheses, rather than seeking to accept the hypothesis which is complementary to the set of falsified hypotheses [53–55]. When generating hypotheses, researchers should consider the inferential framework (e.g. hypothetico-deductive or inductive/probabilistic) to which their hypotheses will ultimately be subjected. Will a 'crucial experiment' [24] be possible or should probabilistic support be evaluated across a set of models (e.g. [45,56])?

Platt [24] has several fine suggestions for conceiving of competing hypotheses, in the context of 'strong inference': dedicated time/effort to the task; using logic trees to describe the system; and modularizing the processes. Betini *et al.* [26] also discuss potential barriers to hypothesis generation and suggest several new approaches to overcome those barriers. Burnham and Anderson [45] also provide much useful guidance for the generation of competing hypotheses under an inductive framework (information theory). We would also add that the entire workflow is iterative, and the construction, implementation and analysis of models may also help to reveal additional hypotheses (e.g. see electronic supplementary material, 2).

## 5.2. Step 2: write a formal model for each hypothesis

In this step, the researcher converts the conceptual models to formal models. There is considerable flexibility in the type of model promulgated here. Researchers could generate fully mechanistic mathematical models or phenomenological statistical models, fully deterministic models or stochastic

**Box 1.** Vetting hypotheses about what drives species distributions.

Species distribution models (SDMs) seek to explain the spatial distribution and abundance of organisms as a function of some environmental variable(s). However, these models often overfit datasets with the complex combinations of environmental variables while failing to provide useful predictive power resulting in occasionally impossible parameter estimates or model selections [36]. For example, Fourcade *et al.* [37] demonstrated that rasterized paintings projected onto the landscape provided comparable or better fitting models for species distributions than real environmental variables (figure 3). This suggests that SDMs may not be considering the full range of potential mechanistic drivers of species distributions (e.g. conspecific attraction, neutral distributions, density dependence, etc.)

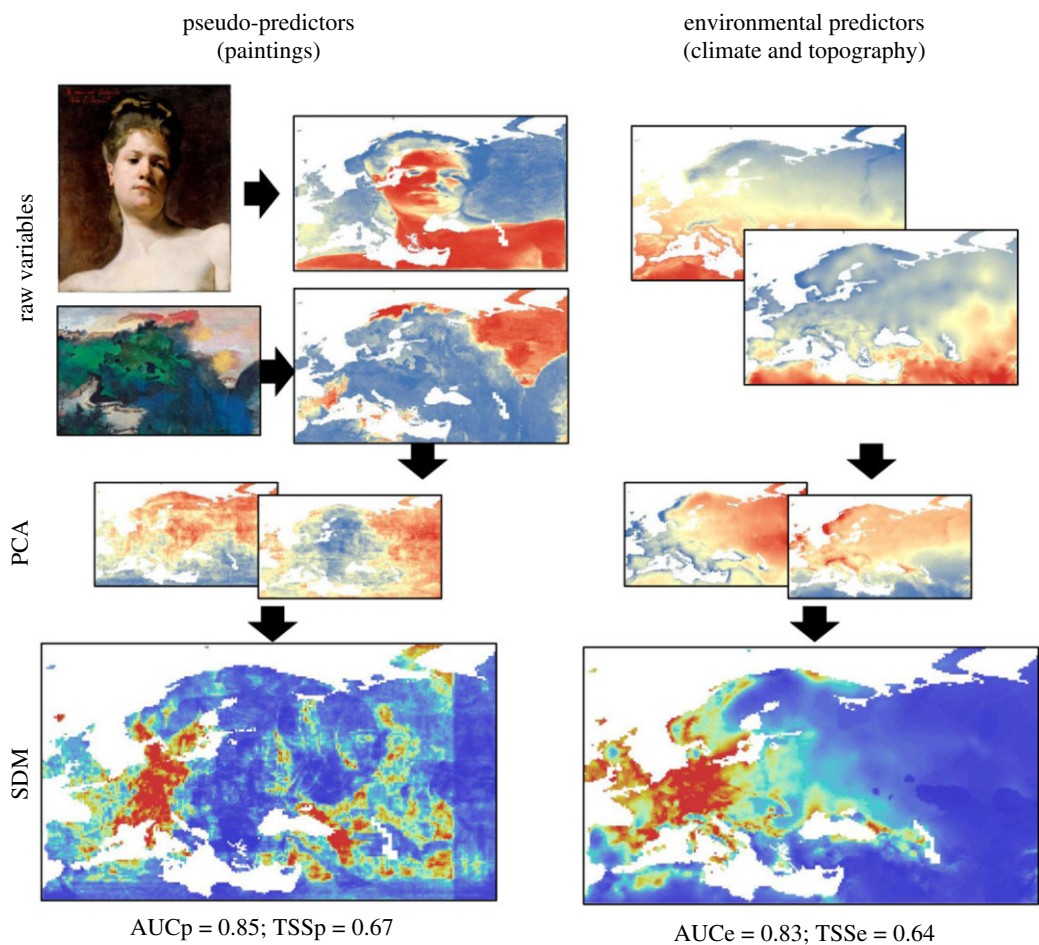

**Figure 3**. Reproduced from [37] 'Workflow used in analyses: 20 pseudo-predictors were created from the projection of paintings on the Western Palaearctic geographical space (examples: top: John Singer Sargent, Blonde Model, bottom: Zhang Daqian, Spring dawns upon the colourful hills) and were used to compute species distribution models (SDMs) after principal components analysis (PCA). A set of 20 true environmental variables (climate and topography) was also used to compute SDMs for the same species. Both types of models were evaluated using area under the receiver operating curve (AUC) and true skill statistics (TSS). The SDMs presented at the bottom show the example of a species (*Candidula unifasciata*, a land snail species) for which the SDM computed with pseudo-predictors led to better evaluation metrics (here computed by randomly splitting occurrences into training and testing datasets) than that computed with real environmental variables (suitability increases from blue to red). AUCp = AUC for model computed with painting-derived pseudo-predictors; AUCe = AUC for model computed with real environmental variables; TSSp = TSS for model computed with painting-derived pseudo-predictors; TSSe = TSS for model computed with real environmental variables.'

We used a simple individual-based simulation model (more details in electronic supplementary material, 1) of animals settling a landscape to consider multiple competing hypotheses about processes that give rise to species distributions. Specifically, we vetted three competing hypotheses about how a population of animals may settle a patchy landscape:

1. *Null model.* Individuals settle the landscape randomly with no influence of habitat or neighbours.
2. *Habitat preference (HP) model.* Individuals settle the landscape preferring 'Habitat A' over 'Habitat B'.
3. *Conspecific attraction (CA) model.* Individuals settle the landscape preferring to settle near already-settled locations.

### Noisy hypotheses

In order to examine the variances produced by each model, compare variances between models and examine how variance relates to parametrization, we calculated and plotted the range for each sampling distribution produced by the 11 model parametrization combinations (figure 4).

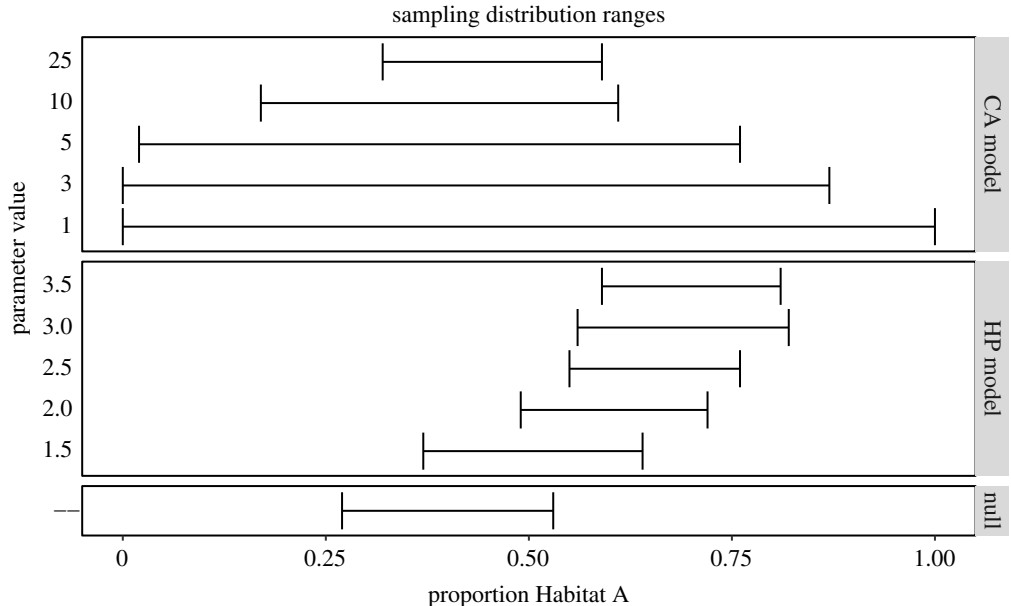

**Figure 4.** Whisker plot of sampling distribution ranges for each parametrization of each hypothesis used to detect noisy hypotheses. Wider whiskers indicate lower precision in parameter estimation and potential evidence of a noisy relationship.

The models containing the strongest conspecific attraction produced the highest variances. As conspecific attraction gets weaker, the values and variance become comparable to the null model. There is also clear structure in the values estimated by the habitat preference models: we observed a higher proportion of 'Habitat A' selected by models containing stronger habitat preference. Variance was relatively constant between models suggesting that parameter estimation under this hypothesis would be similarly accurate regardless of the magnitude of the parameter estimate itself.

### Hypothesis degeneration

To compare sampling distributions to each other to search for degenerate relationships, we calculated the unidirectional pairwise overlap between all sampling distributions. Each overlap was unidirectional, since different model parametrizations produced unequal variances—the overlap between any two sampling distributions was asymmetric. We combined all unidirectional pairwise comparisons into heatmaps to assess patterns of overlap in parameter combinations; each unidirectional pairwise proportion of overlap represents the conditional probability of one hypothesis generating response data capable of being produced by another hypothesis (figure 5).

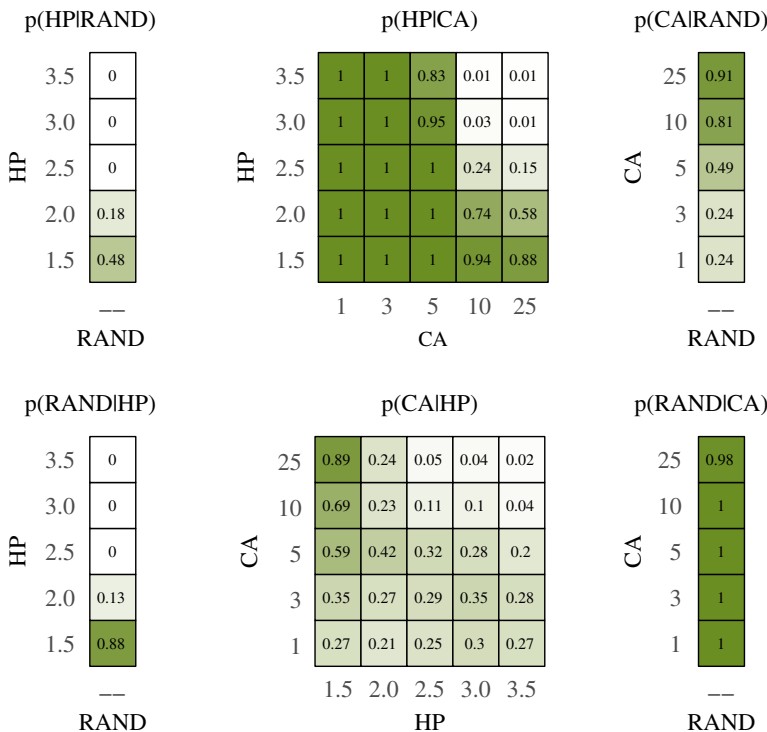

**Figure 5.** Heatmap of sampling distribution overlap. Panels clockwise from top-left: p(HP|RAND) shows the proportion of habitat preference model simulations that overlapped the range of null models for each parametrization; p(HP|CA) shows the proportion of habitat preference model simulations that overlapped the range of conspecific attraction models for each parametrization; p(CA|RAND) shows the proportion of conspecific attraction model simulations that overlapped the range of null models for each parametrization; p(RAND| CA) shows the proportion of null model simulations that overlapped the range of conspecific attraction models for each parametrization; p(CA|HP)shows the proportion of conspecific attraction model simulations that overlapped the range of habitat preference models for each parametrization; p(RAND|HP) shows the proportion of null model simulations that overlapped the range of habitat preference models for each parametrization.

We observed clear structures in the degeneracy of certain model parametrization combinations. For example, the proportion of habitat preference models overlapped by conspecific attraction models was very high for models with low strength of preference and/or strong conspecific attraction. Conversely, the proportion of conspecific attraction models that overlapped habitat preference models was generally low except for models with very strong habitat preference and strong conspecific attraction (figure 5).

### Revising hypotheses

Given both the large variance generated for the null model and the high amount of overlap in sampling distributions between several model parametrization combinations, it is reasonable to assume that a researcher in this situation would seek to refine their proposed study. There are myriad options for such revision and in a 'real-world' examination this would rest on the judgement and system-specific knowledge of the researcher as well as the specific aims of the study. We offer a few potential revisions here to illustrate the types of changes that could be made but in no way suggest that these revisions are exhaustive or appropriate to the system.

By including spatial measures as part of the observed response pattern, models that produced degenerate response patterns may now be parsed. For example, many of the models that hypothesized conspecific attraction exhibited strong spatial clustering, probably resulting from the strong influence of the initially settled location (figure 6).

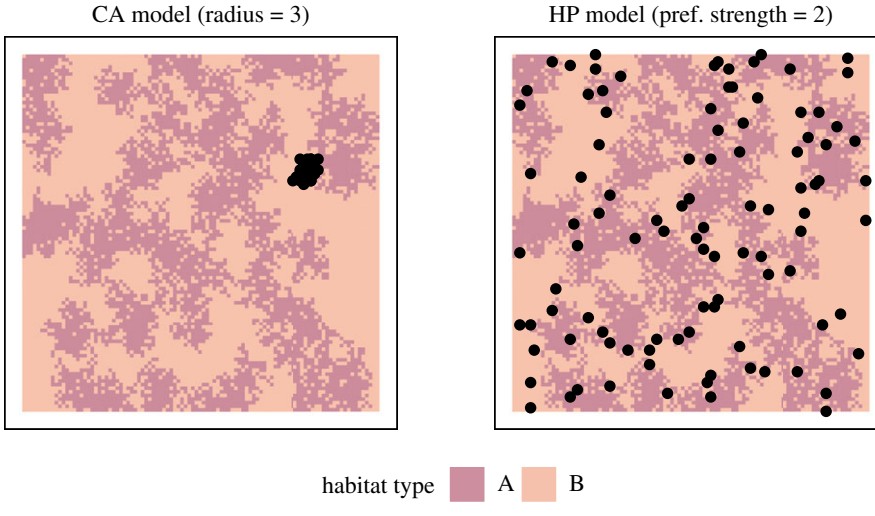

**Figure 6.** Examples of spatial distributions of settled agents in two model iterations. Left: A strong conspecific attraction parametrization. Note the very strong spatial clustering. Right: Parametrized for habitat preference, the model generates a more diffuse spatial pattern. While both these models produced substantially overlapping sampling distributions, spatial metrics could be used to parse hypotheses.

Manipulative experimentation could also parse convergent hypotheses. For example, decoy experiments have been used as a test of conspecific attraction (e.g. [38]). Alternatively, habitat manipulation could also parse degenerate hypotheses (e.g. [39]).

Addressing model parametrizations exhibiting high levels of variance may be more difficult. Because the simulation model assumes no observation error, additional processes or poorly constrained processes are the likely culprits. Indeed, we can see that the spatial distribution in the conspecific attraction model is actually a combination of two separate processes: (i) the initial individual settles randomly and (ii) subsequent individuals settle based on the conspecific attraction decisions rules.

simulations. The chosen model type should account for the complexity of the underlying processes, the relevant level of biological organization under study and the nature of the available data (i.e. do the data allow for direct observation of mechanistic processes?; [57]). Key questions to guide the development of hypotheses should include: what data can actually be collected and does the model match those data? What level(s) of biological organization is relevant to the question and does the model match that hierarchy? Should the hypothesis directly model all relevant processual steps or do latent variables need to be included? Is enough known about the subject to specify a truly process-based model or should statistical links between phenomena be simulated without direct mechanistic components? Is the process deterministic or stochastic?

Seemingly straightforward, this step can be surprisingly complex. Verbal models do not always have obvious mathematical or computer code analogues and creative solutions may be required to translate conceptual models to formal ones. Keep in mind that each version of a model and each parametrization of a version requires its own specification and subsequent analysis, so it pays to think clearly and succinctly about identifying the set of models in step 1.

This step is also the point to consider the logical plausibility of a hypothesis. By formally translating a hypothesis into a model, one is immediately confronted with the logical structure of that hypothesis [22]. At this step, illogical hypotheses reveal themselves and can be corrected or removed from the set of working hypotheses. Note that logical consistency is not equivalent to 'truth'; it is an indication that the hypothesis/model is internally coherent. For example, the intermediate disturbance hypothesis (IDH; [58,59]), as originally stated, contained internally incoherent elements such that the premises of the model did not support the predictions [60]. By specifying the IDH as a mathematical population model, Fox [60] showed that intermediate disturbance frequencies do not, in fact, predict 'hump-shaped' diversity curves. Interestingly, both Fox [60] and Sheil and Burslem [61] point out that modern

competition–colonization trade-off theory (e.g. [62]) rescues the IDH from logical implausibility, exemplifying the model plausibility check for which we advocate here.

## 5.3. Step 3: generate sampling distributions

Because many models in ecology may contain at least some stochastic components [56], the simulated patterns can vary across iterations (where an iteration is 'running' the model once to generate a single simulated pattern). Therefore, a single iteration is insufficient to compare one candidate hypothesis to another—how could we know if the difference between patterns is due to 'real' differences between models/hypotheses or to inter-iteration stochasticity? Just as data-based inference is centred on estimated sampling distributions, hypothesis vetting is centred on sampling distributions derived from multiple iterations of a simulation model (or the direct calculation of a sampling distribution using a closed-form model). With sampling distributions for a set of working hypotheses in hand, a researcher can identify degenerate and/or noisy relationships among the multiple working hypotheses by comparing the sampling distributions.

A sampling distribution is required for all hypotheses under consideration, including any/all parametrization(s) thereof. Thus, for a model containing a single free parameter that may assume a range of values, the researcher must generate a sampling distribution for all such parameter values (or at least a bracketed range of parametrizations). Therefore, careful articulation of plausible parametrizations is recommended, because densely sampling the parameter space (the set of possible values a parameter could take) quickly increases computational burden.

## 5.4. Step 4: quantify overlap between sampling distributions

In this step, simulated sampling distributions of response variables are examined for evidence of degeneracy between or noisiness within hypotheses. This process resembles inference performed with data but in this case the inference is between simulated data from modelled hypotheses. This kind of inference can help differentiate the relative identifiability of hypotheses but does not provide support for or against any one hypothesis itself.

To detect a degenerate relationship, we want to quantify the extent to which the output of one simulated hypothesis could also have been generated by any of the others. Simple comparisons of the proportion of a sampling distribution overlapping some plausibly bracketed range of another model's sampling distribution provides a first order estimate. For example, if the sampling distribution generated by Process A is entirely contained within the sampling distribution generated by Process B, then no observation of a pattern consistent with Process A could ever rule out Process B. Note that this calculation is conceptually equivalent to the familiar $p$-value from the null hypothesis testing framework but can be used to compare the probability of any model output conditional upon any other model (including, but not limited to, a simulated null). To detect a noisy relationship, we simply quantify the variance within a sampling distribution relative to the magnitude of the estimate. Box 1 and electronic supplementary material, 1 and 2 contain examples of quantifying and visualizing both convergent and noisy relationships.

Classifying hypotheses as degenerate or noisy requires context-specific judgement by the researcher. No pre-prescribed degree of overlap between two hypotheses is automatically 'too much', nor is there a standard upper limit for variance. Instead, the researcher must decide if the precision with which parameters may be estimated or hypotheses may be parsed is sufficient for the purposes of answering the question at hand. This judgement requires system-specific knowledge and sober consideration of the ultimate inferential aims (see [63]). Key questions to consider include: what level of precision is required for the estimated parameter? This will depend on e.g. expected effects sizes, intended uses of the research output and the scale at which the ecological process unfolds. What probability of error in hypothesis selection/rejection is acceptable—is this work exploratory or confirmatory? Exploratory work may be more forgiving of a moderate probability of error whereas confirmatory work may be incompatible with all but a very low probability of error.

## 5.5. Step 5: revise hypotheses and repeat vetting procedure

If the results of the previous step indicate that hypotheses are degenerate or noisy, the researcher must consider whether they can be adequately revised while remaining biologically relevant. Degenerate

hypotheses can be replaced by alternative hypotheses, including revisions that more explicitly address problematic confounding issues arising from measurement methods (e.g. studies where detection of an event is imperfect). In other words, researchers might think carefully about sources of variance not included in the hypotheses that would help parse the observable patterns. Modelling allows a researcher to quantify the degree of degeneracy in a set of hypotheses and to test alternative measures or analyses that lead to identifiability.

When the noisy relationship problem is encountered, pattern variance unrelated to the mechanism is often the culprit. In such cases, researchers can consider approaches to either reduce observation error or to better define or constrain the hypothesis about the relationship between process and pattern. Reducing variance from observation or measurement error can be straightforward: improve measurement techniques by design or integrate models for observation error into the analysis. For example, the spatial resolution of modern global positioning system tracking devices is orders of magnitude more precise than, for example, banding data or intrinsic geographic markers such as tissue stable isotopes [64]; occupancy modelling represents a widely used incorporation of error variance into estimates of species distributions [65]. Reducing variance from unconstrained hypotheses requires refining proposed models for the underlying processes. This may be a matter of reducing the stochastic complexity of model structure, or of adding deterministic processes (increasing the complexity) to the hypothesis. Very simple models often sacrifice predictive specificity in seeking broad generality [66,67]. For example, neutral theory of species coexistence [68] predicts highly variable sampling distributions of community compositions. However, increasing model complexity to include niche stabilizing forces [69] improves the model's predictive specificity [70].

It might also be helpful to reconsider the response variable or the study design. For example, extending the response variable from univariate to multivariate might help to specify confounding covariance. Alternatively, manipulative experimentation may help to parse hypotheses (see box 1 for examples of both). If neither modification to the response variable nor manipulative experimentation is likely to solve the problem, it may be necessary to fully revise or reconsider the hypothesis itself. This makes sense when the simulated noisiness or degeneracy results from a model misspecification (i.e. cases where the processes should have been uniquely detectable from pattern but were not). This is often also an indication that some processes have been omitted from the candidate hypothesis set. If hypotheses are revised in any way, either via modification to response variables, proposing experimentation, or revising hypotheses entirely, the hypothesis vetting process is then repeated until a workable set of hypotheses is identified at which point data-based inference proceeds.

# 6. Conclusion

Pre-data collection modelling of multiple hypotheses should be considered the default mode for scientific investigations. Both NHST and multi-model approaches are susceptible to inferential errors when alternative hypotheses are *a priori* non-identifiable and never formally considered. Adopting a multi-hypothesis approach to data-based inference is a necessary but insufficient first step. Ecologists ought to also consider in the abstract (prior to collecting data) whether proposed hypotheses are even theoretically uniquely identifiable. We have outlined a simple framework for determining the identifiability of hypotheses *a priori* by invoking Chamberlin's 130-year-old method of multiple working hypotheses. We hope that wider adoption of this approach will lead to more robust inference in ecology.

Ethics. No human or animal subjects were included in this study. Therefore, no ethical approvals were required.

Data accessibility. No data are included in this paper. All code generating worked examples are included in electronic supplementary material, 1 and 2, and the *checkyourself* package used in those examples is available on github: https://github.com/syanco/checkyourself and has been archived within the Zenodo repository https://doi.org/10.5281/zenodo.3743038.

Authors' contributions. S.W.Y., A.M., L.H. and M.B.W. conceived of the original idea; S.W.Y. and C.N.T. substantially revised the structure and aims of the paper. S.W.Y. wrote the initial draft and all authors contributed critically to revisions. All authors gave final approval for publication.

Competing interests. The authors declare no competing interests.

Funding. S.W.Y. and A.M. were supported by teaching assistantships from the University of Colorado Denver. Collaboration between S.W.Y. and C.N.T. was supported by the Fritz Knopf Fellowship.

Acknowledgements. The authors are grateful to Elizabeth Hobson for helpful comments on an early version.

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
