## [Reviewer comments · Royal Society Open Science]

Review History

RSOS-200231.R0 (Original submission)

Review form: Reviewer 1 (Gustavo Betini)

Is the manuscript scientifically sound in its present form?

Yes

Are the interpretations and conclusions justified by the results?

Yes

Is the language acceptable?

Yes

Do you have any ethical concerns with this paper?

No

Have you any concerns about statistical analyses in this paper?

No

Recommendation?

Major revision is needed (please make suggestions in comments)

Comments to the Author(s)

General comments:

This is a well-written and interesting manuscript addressing the important issue of multiple hypotheses testing in ecology. The strength and novelty of the manuscript is to move the discussion forward with a pragmatic approach (the pre-data collection modelling). However, I believe the authors did not properly separate what is new in their approach and what has already been proposed. This is mainly because the authors did not place the discussion into a broader context, i.e. they failed to mention the abundant literature on the topic. In addition, I believe the authors should better explore the limitations of their approach, e.g. when writing models for competing hypotheses is close to impossible. Are there alternatives? There are a number of papers that were written in response to Platt 1964 claiming that the method of multiple hypotheses cannot be applied to ecology because of the complexity of the field. So, I believe the authors should better address the fact that sometimes, specially when we know very little about a pattern/process, translating hypotheses into models might be almost impossible.

I have read this manuscript with much enthusiasms and I enjoyed thinking about the problem. I hope you find my comments/suggestions useful.

Specific comments:

l. 21 It is not clear why "ecology is at risk of experiencing a similar crisis". After reading the paper, I am not convinced that this is the case.

l. 43 - Can you give a couple of examples of this "replication crises" and explain how you concluded that they were caused primarily by failure to consider "alternative hypotheses in combination with an over-reliance on null hypothesis significance testing"? What about data fabrication? Maybe it would be more precise to say that, disregarding data fabrication, some of these problems could be solved by considering alternative hypotheses.

l. 50 - There are so many good papers dealing with this problem! I think the authors have to mentioned that this is a well-known area of investigation and cite some of these previous studies.

l. 69 - For a reader that does not know the literature, the authors give the impression that this is the first time Chamberlin's ideas has been presented as a solution. Platt (1964) seems particularly relevant to this discussion.

l. 84 - Again, are these your concepts or concepts already available in the literature? You have not cited anyone here, so the reader might think that this categorization is a new contribution from your study.

L. 142 - I do not think you have the data to suggest that. One paper (Fraser et al 2019) is certainly not enough. That been said, your general point is important. We, as ecologists, should always look for better practices. So, my advice is to replace the statement that we are experiencing a "replication crisis" with another one that indicates that regardless of whether or not we are about to experience a crisis, we should always aim for better practices.

l. 199 - I would move these 5 steps to the end of the first part of the text (L.65-77), so that the reader can assess the novelty of your approach at the very beginning of the paper.

l. 220 - Are these mathematical models or statistical models? Translating a hypothesis into a model can be done phenomenologically by using a statistical model. Although not ideal, the statistical model could be useful if the mathematical model is too complex.

Gustavo S. Betini

Review form: Reviewer 2

Is the manuscript scientifically sound in its present form?

No

Are the interpretations and conclusions justified by the results?

No

Is the language acceptable?

No

Do you have any ethical concerns with this paper?

No

Have you any concerns about statistical analyses in this paper?

No

Recommendation?

Major revision is needed (please make suggestions in comments)

Comments to the Author(s)

This paper outlines a current paucity of multi-hypothesis testing in ecology and provides a pre-data collection modelling approach to resolve this. I completely agree that there is a problem outlined by the authors. Indeed, practitioners should consider multiple working hypotheses and be more discerning when accepting their resulting inferences. This paper does provide a nice summary of existing literature and there is a suggestion for a way forward.

However, I am not clear exactly what gap this paper is filling in the literature and the novel in silico recommendation is quite abstract. It would be useful to have more explicit recommendations for users and guiding questions. The distinction between degenerate and noisy is not completely clear. The figure and text could do more work to clarify these concepts.

This paper would benefit from having examples to root these ideas in real systems and a case study in a box taking us through the scientific method with their recommendations in mind. What types of ecological data are the authors imagining? To make this widely applicable the authors need to make these connections a bit more salient for readers. In addition, I would push the authors to be more precise with their language. The writing could be tightened up and checked for consistency. Some examples are below but the ideas should be applied to the entirety of the manuscript:

L 33, L35, 163: "understand" is a word that is often relied on but is not precise. Particularly because this paper takes aim at how ecologists test ideas underlying natural phenomena it is important to be clear with our language.

L86: Figure vs Fig on L97

L199-206: Could be translated into a useful graphic/flowchart

L228-229: This would be a great place to show an example of an logical and illogical hypothesis revealing themselves.

L287: Example of this would be useful

L292: Example of this would be useful

S1 and S2: The supplementary information is excellent the figures and concrete examples from this need to be integrated into the paper. I believe that would improve clarity and make this contribution much more concrete, where the paper as it currently stands is not. The examples are very spatial so I would urge the authors to think of another example that could follow their framework to make it more generally applicable, or I would suggest that in the text it is clear they generated these ideas for spatial ecology but there are areas for expansion.

Decision letter (RSOS-200231.R0)

19-Mar-2020

Dear Mr Yanco,

The editors assigned to your paper ("A modern method of multiple working hypotheses to improve inference in ecology") have now received comments from reviewers. We would like you to revise your paper in accordance with the referee and Associate Editor suggestions which can be found below (not including confidential reports to the Editor). Please note this decision does not guarantee eventual acceptance.

Please submit a copy of your revised paper before 11-Apr-2020. Please note that the revision deadline will expire at 00.00am on this date. If we do not hear from you within this time then it will be assumed that the paper has been withdrawn. In exceptional circumstances, extensions may be possible if agreed with the Editorial Office in advance. We do not allow multiple rounds of revision so we urge you to make every effort to fully address all of the comments at this stage. If deemed necessary by the Editors, your manuscript will be sent back to one or more of the original reviewers for assessment. If the original reviewers are not available, we may invite new reviewers.

- Data accessibility

It is a condition of publication that all supporting data are made available either as supplementary information or preferably in a suitable permanent repository. The data

accessibility section should state where the article's supporting data can be accessed. This section should also include details, where possible of where to access other relevant research materials such as statistical tools, protocols, software etc can be accessed. If the data have been deposited in an external repository this section should list the database, accession number and link to the DOI for all data from the article that have been made publicly available. Data sets that have been deposited in an external repository and have a DOI should also be appropriately cited in the manuscript and included in the reference list.

If you wish to submit your supporting data or code to Dryad (<http://datadryad.org/>), or modify your current submission to dryad, please use the following link:
<http://datadryad.org/submit?journalID=RSOS&manu=RSOS-200231>

- **Competing interests**

- **Authors' contributions**

- **Acknowledgements**

- **Funding statement**

on behalf of Professor Matjaz Perc (Associate Editor) and Pete Smith (Subject Editor)
openscience@royalsociety.org

Comments to Author:

Reviewers' Comments to Author:

Reviewer: 1

Comments to the Author(s)

General comments:

This is a well-written and interesting manuscript addressing the important issue of multiple hypotheses testing in ecology. The strength and novelty of the manuscript is to move the discussion forward with a pragmatic approach (the pre-data collection modelling). However, I believe the authors did not properly separate what is new in their approach and what has already been proposed. This is mainly because the authors did not place the discussion into a broader context, i.e. they failed to mention the abundant literature on the topic. In addition, I believe the authors should better explore the limitations of their approach, e.g. when writing models for competing hypotheses is close to impossible. Are there alternatives? There are a number of papers that were written in response to Platt 1964 claiming that the method of multiple hypotheses cannot be applied to ecology because of the complexity of the field. So, I believe the authors should better address the fact that sometimes, specially when we know very little about a pattern/process, translating hypotheses into models might be almost impossible.

I have read this manuscript with much enthusiasms and I enjoyed thinking about the problem. I hope you find my comments/suggestions useful.

Specific comments:

l. 21 It is not clear why "ecology is at risk of experiencing a similar crisis". After reading the paper, I am not convinced that this is the case.

l. 43 - Can you give a couple of examples of this "replication crises" and explain how you concluded that they were caused primarily by failure to consider "alternative hypotheses in combination with an over-reliance on null hypothesis significance testing"? What about data fabrication? Maybe it would be more precise to say that, disregarding data fabrication, some of these problems could be solved by considering alternative hypotheses.

l. 50 - There are so many good papers dealing with this problem! I think the authors have to mentioned that this is a well-known area of investigation and cite some of these previous studies.

l. 69 - For a reader that does not know the literature, the authors give the impression that this is the first time Chamberlin's ideas has been presented as a solution. Platt (1964) seems particularly relevant to this discussion.

l. 84 - Again, are these your concepts or concepts already available in the literature? You have not cited anyone here, so the reader might think that this categorization is a new contribution from your study.

L. 142 - I do not think you have the data to suggest that. One paper (Fraser et al 2019) is certainly not enough. That been said, your general point is important. We, as ecologists, should always look for better practices. So, my advice is to replace the statement that we are experiencing a "replication crisis" with another one that indicates that regardless of whether or not we are about to experience a crisis, we should always aim for better practices.

l. 199 - I would move these 5 steps to the end of the first part of the text (L.65-77), so that the reader can assess the novelty of your approach at the very beginning of the paper.

l. 220 - Are these mathematical models or statistical models? Translating a hypothesis into a model can be done phenomenologically by using a statistical model. Although not ideal, the statistical model could be useful if the mathematical model is too complex.

Gustavo S. Betini

Reviewer: 2

Comments to the Author(s)

This paper outlines a current paucity of multi-hypothesis testing in ecology and provides a pre-data collection modelling approach to resolve this. I completely agree that there is a problem outlined by the authors. Indeed, practitioners should consider multiple working hypotheses and be more discerning when accepting their resulting inferences. This paper does provide a nice summary of existing literature and there is a suggestion for a way forward.

However, I am not clear exactly what gap this paper is filling in the literature and the novel *in silico* recommendation is quite abstract. It would be useful to have more explicit recommendations for users and guiding questions. The distinction between degenerate and noisy is not completely clear. The figure and text could do more work to clarify these concepts.

This paper would benefit from having examples to root these ideas in real systems and a case study in a box taking us through the scientific method with their recommendations in mind. What types of ecological data are the authors imagining? To make this widely applicable the authors need to make these connections a bit more salient for readers. In addition, I would push the authors to be more precise with their language. The writing could be tightened up and checked for consistency. Some examples are below but the ideas should be applied to the entirety of the manuscript:

L 33, L35, 163: “understand” is a word that is often relied on but is not precise. Particularly because this paper takes aim at how ecologists test ideas underlying natural phenomena it is important to be clear with our language.

L86: Figure vs Fig on L97

L199-206: Could be translated into a useful graphic/flowchart

L228-229: This would be a great place to show an example of an logical and illogical hypothesis revealing themselves.

L287: Example of this would be useful

L292: Example of this would be useful

S1 and S2: The supplementary information is excellent the figures and concrete examples from this need to be integrated into the paper. I believe that would improve clarity and make this contribution much more concrete, where the paper as it currently stands is not. The examples are very spatial so I would urge the authors to think of another example that could follow their framework to make it more generally applicable, or I would suggest that in the text it is clear they generated these ideas for spatial ecology but there are areas for expansion.

Author's Response to Decision Letter for (RSOS-200231.R0)

See Appendix A.

RSOS-200231.R1 (Revision)

Review form: Reviewer 1 (Gustavo Betini)

Is the manuscript scientifically sound in its present form?

Yes

Are the interpretations and conclusions justified by the results?

Yes

Is the language acceptable?

Yes

Do you have any ethical concerns with this paper?

No

Have you any concerns about statistical analyses in this paper?

No

Recommendation?

Accept as is

Comments to the Author(s)

Thank you for taking the time to address my comments. This is a great manuscript and I hope to see it published soon.

Review form: Reviewer 2

Is the manuscript scientifically sound in its present form?

Yes

Are the interpretations and conclusions justified by the results?

Yes

Is the language acceptable?

Yes

Do you have any ethical concerns with this paper?

No

Have you any concerns about statistical analyses in this paper?

No

Recommendation?

Accept with minor revision (please list in comments)

Comments to the Author(s)

Scientists embark on the journey of revealing true processes underlying observed patterns but there has been a considerable hurdle where inferences be an artifact of the practitioner's biases and decisions. This work continues the conversation about our collective responsibility to be conscious of and address possible shortcomings of our research. Particularly, we must test multiple hypotheses that may describe the mechanism underlying our observations. The paper

provides a nice overview/review of this discussion over the decades and presents a contemporary solution to add to our toolkit as scientists, considering multiple hypotheses before data collection.

I reviewed an earlier version of this manuscript and I am very content with the revisions. The authors addressed all concerns and suggestions thoughtfully. I appreciate the effort the authors put in to these revisions.

I have no major comments. Their work will make a valuable contribution.

Minutiae (None of these are critical to address)

L34: what is "true"? Perhaps just "discover possible relationships"

L87: Is the list of intentions brought up later? I think having two numbered lists might be a bit confusing especially so close together in your conclusions. Maybe labelling this list as Goal 1 and then the Steps starting on line 94 as Step 1 etc. if you refer to them.

Do the sections have to be numbered? I am trying to compare them to the other numbered lists. I know the numbering of headings may be based on journal format but it would be nice if only numbers were used intentionally and for repeatedly references lists.

L127: Remove 'Again' since it is re-visited implying another visit unless this is the third visit

L148: How does the researcher figure out how many hypotheses are plausible?

L246: "hypothesis; As the falsification" should be a period or not capitalized?

L340-341: Could the authors discuss these two questions like they do the third?

L511: Betini et al 2017 is sometimes 2016 – please go through this and other references for consistency

Decision letter (RSOS-200231.R1)

Dear Mr Yanco:

On behalf of the Editors, I am pleased to inform you that your Manuscript RSOS-200231.R1 entitled "A modern method of multiple working hypotheses to improve inference in ecology" has been accepted for publication in Royal Society Open Science subject to minor revision in accordance with the referee suggestions. Please find the referees' comments at the end of this email.

The reviewers and Subject Editor have recommended publication, but also suggest some minor revisions to your manuscript. Therefore, I invite you to respond to the comments and revise your manuscript.

- Ethics statement

If your study uses humans or animals please include details of the ethical approval received, including the name of the committee that granted approval. For human studies please also detail

whether informed consent was obtained. For field studies on animals please include details of all permissions, licences and/or approvals granted to carry out the fieldwork.

- Data accessibility

If you wish to submit your supporting data or code to Dryad (<http://datadryad.org/>), or modify your current submission to dryad, please use the following link:
<http://datadryad.org/submit?journalID=RSOS&manu=RSOS-200231.R1>

- Competing interests

- Authors' contributions

- Acknowledgements

- Funding statement

Because the schedule for publication is very tight, it is a condition of publication that you submit the revised version of your manuscript before 16-May-2020. Please note that the revision deadline will expire at 00.00am on this date. If you do not think you will be able to meet this date please let me know immediately.

on behalf of Professor Matjaz Perc (Associate Editor) and Pete Smith (Subject Editor)
openscience@royalsociety.org

Reviewer comments to Author:
Reviewer: 1

Comments to the Author(s)
Thank you for taking the time to address my comments. This is a great manuscript and I hope to see it published soon.

Reviewer: 2

Comments to the Author(s)

Scientists embark on the journey of revealing true processes underlying observed patterns but there has been a considerable hurdle where inferences be an artifact of the practitioner's biases and decisions. This work continues the conversation about our collective responsibility to be conscious of and address possible shortcomings of our research. Particularly, we must test multiple hypotheses that may describe the mechanism underlying our observations. The paper provides a nice overview/review of this discussion over the decades and presents a contemporary solution to add to our toolkit as scientists, considering multiple hypotheses before data collection.

I reviewed an earlier version of this manuscript and I am very content with the revisions. The authors addressed all concerns and suggestions thoughtfully. I appreciate the effort the authors put in to these revisions.

I have no major comments. Their work will make a valuable contribution.

Minutiae (None of these are critical to address)

L34: what is "true"? Perhaps just "discover possible relationships"

L87: Is the list of intentions brought up later? I think having two numbered lists might be a bit confusing especially so close together in your conclusions. Maybe labelling this list as Goal 1 and then the Steps starting on line 94 as Step 1 etc. if you refer to them.

Do the sections have to be numbered? I am trying to compare them to the other numbered lists. I know the numbering of headings may be based on journal format but it would be nice if only numbers were used intentionally and for repeatedly references lists.

L127: Remove 'Again' since it is re-visited implying another visit unless this is the third visit

L148: How does the researcher figure out how many hypotheses are plausible?

L246: "hypothesis; As the falsification" should be a period or not capitalized?

L340-341: Could the authors discuss these two questions like they do the third?

L511: Betini et al 2017 is sometimes 2016 - please go through this and other references for consistency

Author's Response to Decision Letter for (RSOS-200231.R1)

See Appendix B.

Decision letter (RSOS-200231.R2)

Dear Mr Yanco,

It is a pleasure to accept your manuscript entitled "A modern method of multiple working hypotheses to improve inference in ecology" in its current form for publication in Royal Society Open Science.

Kind regards,
Lianne Parkhouse
Royal Society Open Science
openscience@royalsociety.org

on behalf of Professor Matjaz Perc (Associate Editor) and Pete Smith (Subject Editor)
openscience@royalsociety.org

Appendix A

1 REVIEW

**A modern method of multiple working hypotheses to improve inference in ecology**

Scott W. Yanco^{1,*}, Andrew Mcdevitt¹, Clive N. Trueman², Laurel Hartley¹, Michael B. Wunder¹

RUNNING TITLE: Multiple hypotheses to improve inference

¹Department of Integrative Biology, University of Colorado Denver, Denver, CO, USA

²Ocean and Earth Science, University of Southampton, National Oceanography Centre,

Southampton, UK

*corresponding author: scott.yanco@ucdenver.edu

Formatted: Normal

**ABSTRACT**

Science provides a method to ~~understand~~ learn about the relationships between observed
patterns and the processes that generate them. However, inference can be confounded when an
observed pattern cannot be clearly and wholly attributed to a hypothesized process. Over-
reliance on traditional single-hypothesis methods (i.e., null hypothesis significance testing) has
resulted in replication crises in several disciplines; ~~and ecology is at risk of experiencing a~~
similar crisis exhibits features common to these fields (e.g., low-power study designs,
questionable research practices, etc.). Considering multiple working hypotheses in combination
with pre-data collection modeling can be an effective means to mitigate many of these problems.
We present a framework for ~~understanding~~ explicitly modeling systems in which relevant
processes are commonly omitted, overlooked, or not considered and provide a formal workflow
for a pre-data collection analysis of multiple candidate hypotheses. We advocate for and suggest
ways that pre-data collection modeling can be combined with consideration of multiple working
hypotheses to improve the efficiency and accuracy of research in ecology.

**KEYWORDS**

multiple hypotheses, simulation models, modeling, inference, scientific method

**1 REPLICATION CRISES AND INFERENTIAL FRAMEWORKS**

The ultimate goal of science is to ~~understand~~ learn about the true relationships between
observable patterns in the world around us and the processes that generate those patterns. Most
commonly, scientists ~~come to understand~~ identify and/or quantify the what links between process
~~to and~~ pattern by hypothesizing the existence of a particular relationship between the two and
using data as evidence for or against that hypothesis. However, inference may be unreliable if the
scientist does not consider all potentially relevant processes. For example, inference is
confounded when unconsidered hypotheses produce the same observed pattern as the stated
hypothesis. Similarly, inference is muddled when hypotheses overlook additional variance-
inflating processes, effectively rendering the link between process and pattern
~~undetectable~~ indiscernible. In either case, researchers who do not carefully guard against such
pitfalls may make inferences that are either too strong or too weak.

Recently, several scientific disciplines have experienced “replication crises” (e.g., cancer
biology [Begley and Ellis 2012] and psychology [Open Science Collaboration 2015] among
others [McNutt 2014]). ~~While m~~ Many factors have ~~certainly likely~~ contributed to replication
crises (~~e.g.,~~ publication bias ~~{[Nissen et al. 2016, Greenwald 1975],}~~), hypothesizing after results
are known ~~{[Kerr 1998]}~~, p-hacking ~~{[Bruns and Ioannidis 2016], Greenwald 1992}~~, and data
fabrication (Fanelli 2009) to name a few ~~etc.~~; ~~the primary~~ In addition to these factors,
inferential failure in these disciplines was the presence of unconsidered alternative hypotheses in
combination with an irreproducibility has also been driven by an over-reliance on null hypothesis
significance testing (NHST; ~~which is ill-equipped to handle such circumstances~~ (Ioannidis
2005; Tobias 2011; Begley and Ellis 2012; Button *et al.* 2013; Wasserstein and Lazar 2016). The
limitations, misuse, and outright abuse of NHST are myriad and, by now, well known (see, for

example, Greenwald 1975, Anderson et al. 2000, Greenland et al. 2016, Wasserstein and Lazar
2016, Amrhein et al. 2019). NHST produces erroneous inference both because it is frequently
misinterpreted by researchers (Greenland et al. 2016, Wasserstein and Lazar 2016, Colquhoun
2017, Lash 2017, Loken and Gelman 2017) and because it is prone to manipulation (Greenwald
1975, Greenland et al. 2016).

One potentially underappreciated limitation of NHST is that it does not produce
evidential support for hypotheses, instead providing only weak evidence of incongruence
between observed data and a null hypothesis (Wasserstein and Lazar 2016). The ubiquitous p-
value quantifies only the probability of hypothetical future data resulting in some summary
statistic that would be less consistent with summary statistics computed from data generated by
the null hypothesis. If that probability is sufficiently low (e.g., $p < 0.05$), the researcher “rejects”
the null hypothesis as having been unlikely to generate the observed data (as in Neyman and
Pearson 1933). Often, “rejection of the null” leads (illogically) to acceptance of whatever was
proposed as the alternative to that strawman; the alternative hypothesis is accepted without any
positive inferential support (Greenland et al. 2016). Furthermore, the NHST framework
considers only a single hypothesis. Indeed, the complement to the null hypothesis comprises a *set*
of alternative hypotheses. In other words, a “significant” significance test indicates that data like
ours is improbable given a single null hypothesis (Greenland et al 2016, Wasserstein and Lazar
2016) – it produces no information about the infinite number of possible alternative hypotheses
(Ellison et al 2014). Imagine the potential for error when an automatically-accepted alternative
hypothesis is not uniquely distinguishable from some other hypothesis the researcher never
considered.

Here, we ~~provide an extension to describe~~ methods for considering multiple hypotheses
by advocating for the implementation of multi-hypothesis modeling *prior* to data collection.
Akin to *in silico* experimentation, design phase modeling helps to identify a plausible set of
candidate hypotheses and determine which of the set might ~~plausibly~~ lead to any of several
different observable patterns (Caswell 1988, Servedio et al. 2014). Below, we detail the nature of
problematic sets of hypotheses and ~~present draw on the oft-invoked~~ “method of multiple working
hypotheses” (Chamberlin 1890) as a partial solution. This method has been repeatedly invoked
as an important component of good scientific practice (e.g., Platt 1964, Elliot and Brook 2007,
Betini et al. 2017). ~~We then~~ In this paper we propose a workflow ~~for~~ invoking the method of
multiple working hypotheses in the context of pre-data collection modeling. Our workflow
applies recommended practices in theoretical modeling to the problem of design phase modeling
with particular emphasis on the consideration of multiple hypotheses. The practical
recommendations in our approach is-are intended to 1) facilitate wider adoption of multiple
hypothesis methods; 2) guard against inferential errors to which multi-hypothesis methods are
still prone; and 3) provide a formal framework for such analyses. This combination of multi-
hypothesis inference and pre-data collection modeling represents a powerful alternative
incarnation of the scientific method geared towards stronger inference that is less susceptible to
errors arising from unconsidered processes.

Specifically, we outline five steps for vetting hypotheses. These steps can be repeated
iteratively until the proposed mechanisms and observation patterns adequately map to one
another (Figure 1). The steps are:

- 1. Specify candidate hypotheses;
- 2. Write a model for each hypothesis;

3. Generate sampling distributions of simulated data from each hypothesis:

4. Quantify the variance within and overlap between sampling distributions; and,

5. Revise hypotheses as necessary and repeat steps 1-4.

2 THE EFFECTS OF UNCONSIDERED ALTERNATIVE HYPOTHESES

Scientific inference and, in particular, inference using NHST assumes that processes are
uniquely identifiable from the ~~particular~~ observable patterns they generate (Figure 24). That is,
they depend on the statistical concept of identifiability. Model parameterizations are identifiable
if and only if distinct parameterizations lead to different probability distribution functions
(Casella and Berger 2002).

Muddled inference (i.e. non-identifiability) manifests in two ways: 1) Degenerate
Relationship: multiple processes produce indistinguishable patterns; or, 2) Noisy Relationship:
processes do not reliably produce a single identifiable pattern (Figure 42; Koopmans and
Reiersol 1950, Rothenberg 1971).

2.1 Degenerate Relationship

Hypotheses with degenerate relationships between pattern and process are not testable – a
fundamental requirement to differentiate hypotheses. In degenerate cases, a single observed
pattern could have been produced by more than one process (Figure 42; Koopmans and Reiersol
1950, Rothenberg 1971). Thus, no single process can be uniquely ~~identified from~~ implicated by
the observation. Degeneracy may occur because unconsidered deterministic or stochastic
processes modify the resultant pattern. At its heart, this phenomenon arises due to model
misspecification wherein two or more models (hypotheses), as specified by the researchers,
produce indistinguishable response patterns (Rothenberg 1971). In this situation no observation

Formatted: List Paragraph, Outline numbered + Level: 1 + Numbering Style: 1, 2, 3, ... + Start at: 1 + Alignment: Left + Aligned at: 1.5 pi + Indent at: 3 pi

can serve as evidence of any unique process because multiple processes could have produced the
same pattern.

2.2 Noisy Relationship

Noisy relationships are those wherein a single mechanism produces multiple and varied
response patterns potentially due to unrecognized or unconsidered mechanisms (Figure 42). Too
much variance leads to low predictive power and imprecise estimates of model parameters
(Casella and Berger 2002). ~~This problem is the converse of the Like the~~ degenerate relationship
problem, ~~but this~~ also results in the same muddled inference. Noisy relationships between
patterns and processes commonly arise from observation or measurement errors, or from a
~~comparably uninformed model conceptualization~~ misspecified model.

3 THE METHOD OF MULTIPLE WORKING HYPOTHESES REVISITED AGAIN

While inferential failures leading to replication crises have garnered much recent
attention (Amrhein *et al.* 2019), they are hardly new. Cohen (1994) pointed out flaws in NHST
over twenty-five years ago – and in so doing reminded readers that Bakan (1966) made similar
arguments over thirty years prior to that. In 1964, Platt described “strong inference” which
grounded much of what Ioannidis (2005) demonstrated over forty years later. In fact, as early as
1890, Thomas Chamberlin described the “method of multiple working hypotheses” and it has
since been repeatedly advocated as a way to mitigate the risk of omitting potentially relevant
processes from inference (Chamberlin 1890; Platt 1964; Sagan 1995; Elliott and Brook 2007).

Chamberlin (1890) advocated that, to avoid foreseeable inferential errors, researchers
should explicitly consider *multiple* working hypotheses from the outset. The method is intended
to reduce cognitive biases which cause researchers to only collect evidence for favored
hypotheses. Additionally, Chamberlin points out that single hypothesis frameworks fail to

adequately account for complex systems wherein multiple processes may play causal roles — as
is common in ecology (Chamberlin 1890; Elliott and Brook 2007). Using this method, a
researcher “competes” evidence about as many hypotheses as are plausible rather than simply
considering the evidence against a strawman hypothesis (as in NHST). ~~To many, such an~~
~~approach feels intuitive, and yet adoption of the method still lags.~~

Despite ~~nearly at least~~ 130 years of advocacy for multi-hypothesis approaches,
consideration of multiple hypotheses in ecology continues to be rare (Stephens *et al.* 2007). For
example, Betini *et al.* (2017) found that only 21% of a sample of recently published papers in
ecology and evolution considered multiple hypotheses. Yet, the systems investigated in these
fields are precisely those ~~that which~~ stand to benefit from multi-hypothesis approaches (i.e.,
those involving multiple interacting causal factors; Hilborn and Mangel 1997).

Observable patterns arising from myriad interacting variance-generating processes is the
norm in ecology. Such complex causal structures are prone to both the degenerate and noisy
relationship problems (Hilborn and Mangel 1997). Consider just a few examples chosen from
subdisciplines within ecology: Boeklen *et al.* (2011) identified at least 44, hierarchically
organized, factors that influence emergent patterns of tissue stable isotopes used in trophic
ecology studies. ~~Nathan *et al.* (2008) showed that animal movements emerge from a complex~~
~~interaction between the organism’s motility, capacity to navigate, internal state, and external~~
~~environmental setting — each component of which may themselves entail multiple interacting~~
~~variables (see also Supplement 2). Finally, s~~Several authors have observed sufficient variance in
species distributions to produce absurd or impossible model fits (Bahn and McGill 2007); e.g.,
For example, Fourcade *et al.* (Fourcade *et al.* 2018) demonstrated that rasterized paintings
projected onto landscapes provided comparable or better fitting models for species distributions

than real environmental variables; (see also Box 1 and Supplement 1). Finally, Nathan et al.
(2008) showed that animal movements emerge from an interaction between the organism's
motility, capacity to navigate, internal state, and external environmental setting - each
component of which may themselves entail multiple interacting variables (see Supplement 2).
These are examples of fields wherein identifying mechanistic drivers via observed patterns is
challenging because of the multi-faceted nature of the problems at hand — a ubiquitous scenario
in ecology. As such, establishing the identifiability of the set of plausible hypotheses should be
regarded as the default first step towards reliable inference in ecology.

Compounding the effects of underlying complexity, problematic inferential practices may
be common in ecology displays several features common to fields experiencing substantial
replication failures. For example, Fraser et al. (2017) found that questionable research practices
were widespread in ecology— observing rates comparable to fields whose replication crises are
well established. In fact, recent, large-scale studies have already begun to show that early, low-
power findings in some sub-fields apparently do not replicate (e.g., Clark et al 2020). In
combination, these facts make clear that ecology must improve its inferential toolbox.

Frameworks that support data-based inference between multiple hypotheses are well
established (e.g., information theoretic approaches to multi-model inference; Burnham and
Anderson 1998, 2001, 2002; Anderson *et al.* 2000; Stephens *et al.* 2007; Burnham *et al.* 2011;
Nichols *et al.* 2019), and have even been explicitly linked to Chamberlin's method (Elliott and
Brook 2007). Yet, while some sub-disciplines within ecology have seen wider adoption of these
tools (Elliott and Brook 2007), clearly they remain under-utilized (Stephens *et al.* 2007; Betini *et*
*al.* 2017). More importantly, *a posteriori* multi-hypothesis methods cannot disentangle
hypotheses that are confounded *a priori* (e.g., i.e., those that are structurally are non-

identifiable). As such, ~~any effective approach to inference must employ~~ing Chamberlin's
method at the earliest stages of research may improve inference. Therefore, pre-data collection
modeling is an essential first step in considering multiple hypotheses.

**4 PRE-DATA COLLECTION MODELING ENABLES THE METHOD OF** 196 **MULTIPLE WORKING HYPOTHESES**

Models constructed prior to data collection can provide insights allowing researchers to
~~understand~~, quantify, and, ultimately, to increase clarity and transparency about hypotheses
(Haldane 1964, Hillis 1993, Servedio et al. 2014). These models are essentially *in silico*
experiments which simulate response variables using predefined parameters taking on
biologically relevant values. Specifying a model forces the researcher to explicitly consider the
nature of linkages between the process(es) under investigation and the pattern(s) observed (Hillis
1993). By using biologically defined parameters, the simulated pattern is clearly understood
because the structural components of the model explicitly represent biological links between
process and pattern (Servedio et al. 2014). Comparing simulated responses across multiple
alternative hypotheses allows a researcher to quantify the identifiability of each candidate model.

This step, though formally distinct, is analogous to a power analysis wherein researchers
use pre-data collection models to ensure that the proposed sample will be sufficient to answer the
question at hand. Whereas a power analysis assesses the sufficiency of sample sizes (given some
assumed effect size), our framework assesses the ~~fundamental~~-identifiability of each hypothesis.
Both analyses are ways to ensure, at the outset, that a proposed study is even theoretically
capable of producing an answer.

Modeling in this context embraces the method of multiple hypotheses: researchers
consider not only a favored hypothesis but also alternative formulations. This ~~identifies-uncovers~~

[revised manuscript text omitted]

This step is also the point to consider the logical plausibility of a hypothesis. By formally
~~connecting a model to translating~~ a hypothesis into a model, one is immediately confronted with

the logical structure of ~~the-that~~ hypothesis (Servedio *et al.* 2014). At this step, illogical
hypotheses reveal themselves and can be corrected or removed from the set of working
hypotheses. Note that logical consistency is not equivalent to “truth”; it is an indication that the
hypothesis/model is internally coherent. For example, the intermediate disturbance hypothesis
(IDH; Hutchinson 1961, Connell 1978), as originally stated, contained internally incoherent
elements such that the premises of the model did not support the predictions (Fox 2013). By
specifying the IDH as a mathematical population model, Fox (2013) showed that intermediate
disturbance frequencies do not, in fact, predict “hump-shaped” diversity curves. Interestingly,
both Fox (2013) and Sheil and Burslem (2013) point out that modern competition-colonization
tradeoff theory (e.g., Kondoh 2001) rescues the IDH from logical implausibility, exemplifying
the model plausibility check for which we advocate here.

5.3 Step 3: Generate Sampling Distributions

Because ~~most-many~~ models in ecology ~~will likely may~~ contain at least some stochastic

[revised manuscript text omitted]

Formatted: Font: Bold

Formatted: List Paragraph, Numbered + Level: 1 +
Numbering Style: 1, 2, 3, ... + Start at: 1 + Alignment:
Left + Aligned at: 4.5 pi + Indent at: 6 pi

Formatted: Font: Bold

Formatted: Font: Bold

the null model. There is also clear structure in the values estimated by the habitat preference
models: we observed a higher proportion of “Habitat A” selected by models containing stronger
habitat preference. Variance was relatively constant between models suggesting that parameter
estimation under this hypothesis would be similarly accurate regardless of the magnitude of the
parameter estimate itself.

**Hypothesis Degeneration**

To compare sampling distributions to each other to search for degenerate relationships,
we calculated the unidirectional pairwise overlap between all sampling distributions. Each
overlap was unidirectional since different model parameterizations produced unequal variances –
the overlap between any two sampling distributions was asymmetric. We combined all
unidirectional pairwise comparisons into heatmaps to assess patterns of overlap in parameter
combinations; Each unidirectional pairwise proportion of overlap represents the conditional
probability of one hypothesis generating response data capable of being produced by another
hypothesis (Figure 5).

We observed clear structures in the degeneracy of certain model-parameterization
combinations. For example, the proportion of habitat preference models overlapped by
conspecific attraction models was very high for models with low strength of preference and/or
strong conspecific attraction. Conversely, the proportion of conspecific attraction models that
overlapped habitat preference models was generally low except for models with very strong
habitat preference and strong conspecific attraction (Figure 5).

**Revising Hypotheses**

Given both the large variance generated for the null model and the high amount of
overlap in sampling distributions between several model-parameterization combinations, it’s

reasonable to assume that a researcher in this situation would seek to refine their proposed study.
There are myriad options for such revision and in a “real world” examination this would rest on
the judgement and system-specific knowledge of the researcher as well as the specific aims of
the study. We offer a few potential revisions here to illustrate the types of changes that could be
made but in no way suggest that these revisions are exhaustive or appropriate to the system.

By including spatial measures as part of the observed response pattern, models that
produced degenerate response patterns may now be parsed. For example, many of the models
that hypothesized conspecific attraction exhibited strong spatial clustering, likely resulting from
the strong influence of the initially settled location (Figure 6).

Manipulative experimentation could also parse convergent hypotheses. For example,
decoy experiments have been used as a test of conspecific attraction (e.g., Ward et al. 2011).
Alternatively, habitat manipulation could also degenerate hypotheses (e.g., Cruz-Angón et al.
2008).

Addressing model-parameterizations exhibiting high levels of variance may be more
difficult. Because the simulation model assumes no observation error, additional processes or
poorly constrained processes are the likely culprits. Indeed, we can see that the spatial
distribution in the conspecific attraction model is actually a combination of two separate
processes: 1) the initial individual settles randomly; and 2) subsequent individuals settle based on
the conspecific attraction decisions rules.

**Data Accessibility**

No data are included in this paper. All code generating worked examples are included in
Supplements 1 and 2 and the *checkyourself* package used in those examples is available on

Formatted: Space After: 0 pt, Line spacing: Double

Formatted: Font: Not Italic

[github: https://github.com/syanco/checkyourself](https://github.com/syanco/checkyourself) and has been archived within the Zenodo
repository <https://doi.org/10.5281/zenodo.3743038>.

**Competing Interests**

The authors declare no competing interests.

**Author Contributions**

SY, AM, LH, and MW conceived of the original idea; SY and CT substantially revised the
structure and aims of the paper. SY wrote the initial draft and all authors contributed critically to
revisions. All authors gave final approval for publication.

**Funding Statement**

SY and AM were supported by teaching assistantships from the University of Colorado Denver.

Formatted: Normal

Collaboration between SY and CT was supported by the Fritz Knopf Fellowship.

Formatted: English (United States)

**8 — LITERATURE CITED**

Adler PB, Hillerislambers J, Levine JM. 2007 A niche for neutrality. *Ecol. Lett.* **10**, 95–104.

Amrhein V, Greenland S, and McShane B. 2019. Scientists rise up against statistical
significance. *Nature* **567**: 305–7.

Anderson DR, Burnham KP, and Thompson WL. 2000. Null Hypothesis Testing: Problems,
Prevalence, and an Alternative. *J Wildl Manage* **64**: 912–23.

Bahn V and McGill BJ. 2007. Can niche- based distribution models outperform spatial
interpolation? *Glob Ecol Biogeogr* **16**: 733–42.

Bakan D. 1966. The test of significance in psychological research. *Psychol Bull* **66**: 423–37.

Begley CG and Ellis LM. 2012. Raise standards for preclinical cancer research. *Nature* **483**:
531–3.

Bell G. 2000 The Distribution of Abundance in Neutral Communities. *Am. Nat.* **155**, 606–617.

Betini GS, Avgar T, and Fryxell JM. 2017. Why are we not evaluating multiple competing
hypotheses in ecology and evolution? *Roy Soc Open Sci* **4**: 160756.

Boecklen WJ, Yarnes CT, Cook BA, and James AC. 2011. On the Use of Stable Isotopes in
Trophic Ecology. *Annu Rev Ecol Evol Syst* **42**: 411–40.

Bruns SB and Ioannidis JPA. 2016. p-Curve and p-Hacking in Observational Research. *PLoS*
*One* **11**: e0149144.

Burnham KP and Anderson DR. 1998. Practical Use of the Information-Theoretic Approach. In:
Burnham KP, Anderson DR (Eds). *Model Selection and Inference: A Practical*
*Information-Theoretic Approach*. New York, NY: Springer New York.

Burnham KP and Anderson DR. 2001. Kullback-Leibler information as a basis for strong
inference in ecological studies. *Wildl Res* **28**: 111–9.

Burnham KP and Anderson DR. 2002. *Model Selection and Multimodel Inference*.

Burnham K, Anderson D, and Huyvaert K. 2011. AIC model selection and multimodel inference
in behavioral ecology: some background, observations, and comparisons. *Behav Ecol*
*Sociobiol* **65**: 23–35.

Button KS, Ioannidis JPA, Mokrysz C, *et al.* 2013. Power failure: why small sample size
undermines the reliability of neuroscience. *Nat Rev Neurosci* **14**: 365–76.

Casella G and Berger RL. 2002. *Statistical inference*. Duxbury Pacific Grove, CA.

Caswell, H. 1988. Theory and models in ecology: A different perspective. *Ecological modelling*
43:33–44.

- Chamberlin TC. 1890. The method of multiple working hypotheses. *Science* **15**: 92–6.
- Chesson P. 2000 Mechanisms of Maintenance of Species Diversity. *Annu. Rev. Ecol. Syst.* **31**,
343–366.
- Clark, J. S., and A. E. Gelfand. 2006. A future for models and data in environmental science.
Trends in ecology & evolution 21:375–380.
- Clark, T. D., G. D. Raby, D. G. Roche, S. A. Binning, B. Speers-Roesch, F. Jutfelt, and J.
Sundin. 2020. Ocean acidification does not impair the behaviour of coral reef fishes.
*Nature* 577:370–375.
- Cohen J. 1994. The earth is round ($p < .05$). *Am Psychol* **49**: 997.
- Colquhoun, D. 2017. The reproducibility of research and the misinterpretation of p-values. Royal
Society open science 4:171085.
- Connell JH. 1978 Diversity in tropical rain forests and coral reefs. *Science* **199**, 1302–1310.
- Cruz-Angón A, Sillett TS, Greenberg R. 2008 An experimental study of habitat selection by
birds in a coffee plantation. *Ecology* **89**, 921–927.
- De Veaux, R. D., and P. F. Velleman. 2008. Math is music; statistics is literature (or, why are
there no six-year-old novelists?). *Amstat News* 375:54–58.
- Elliott LP and Brook BW. 2007. Revisiting Chamberlin: Multiple Working Hypotheses for the
21st Century. **57**: 608–14.
- Ellison, A. M., N. J. Gotelli, B. D. Inouye, and D. R. Strong. 2014. P values, hypothesis testing,
and model selection: it's déjà vu all over again. *Ecology* 95:609–610.
- Evans MR et al. 2013 Do simple models lead to generality in ecology? *Trends Ecol. Evol.* **28**,
578–583.
- Fagan WF et al. 2013 Spatial memory and animal movement. **16**, 1316–1329.

Fanelli, D. 2009. How many scientists fabricate and falsify research? A systematic review and
meta-analysis of survey data. PloS one 4:e5738.

Fourcade Y, Besnard AG, and Secondi J. 2018. Paintings predict the distribution of species, or
the challenge of selecting environmental predictors and evaluation statistics. *Glob Ecol*
*Biogeogr* **27**: 245–56.

Fox JW. 2013 The intermediate disturbance hypothesis should be abandoned. *Trends Ecol. Evol.*
28, 86–92.

Fraser H, Parker T, Nakagawa S, *et al.* 2017. Questionable Research Practices in Ecology and
Evolution.

Greenland, S., S. J. Senn, K. J. Rothman, J. B. Carlin, C. Poole, S. N. Goodman, and D. G.
Altman. 2016. Statistical tests, P values, confidence intervals, and power: a guide to
misinterpretations. *European journal of epidemiology* 31:337–350.

Haldane, J. B. 1964. A defense of beanbag genetics. *Perspectives in biology and medicine*
7:343–359.

Hartig F, Calabrese JM, Reineking B, *et al.* 2011. Statistical inference for stochastic simulation
models – theory and application. **14**: 816–27.

Hilborn R and Mangel M. 1997. *The Ecological Detective: Confronting Models with Data.*
Princeton University Press.

Hillis, W. D. 1993. Why physicists like models and why biologists should. *Current biology: CB*
3:79–81.

Holling CS. 1966 The strategy of building models of complex ecological systems. *Systems*
analysis in ecology . 195–214.

Hutchinson GE. 1961 The paradox of the plankton. *Am. Nat.* **95**, 137–145.

Ioannidis JPA. 2005. Why most published research findings are false. *PLoS Med* **2**: e124.

Kays R, Crofoot MC, Jetz W, Wikelski M. 2015 Terrestrial animal tracking as an eye on life and
planet. *Science* **348**, aaa2478.

Kerr NL. 1998. HARKing: hypothesizing after the results are known. *Pers Soc Psychol Rev* **2**:
196–217.

Kondoh M. 2001 Unifying the relationships of species richness to productivity and disturbance.
*Proc. Biol. Sci.* **268**, 269–271.

Koopmans, T. C., and O. Reiersol 1950. The Identification of Structural Characteristics. *Annals*
of Mathematical Statistics 21:165–181.

Lash, T. L. 2017. The Harm Done to Reproducibility by the Culture of Null Hypothesis
Significance Testing. *American journal of epidemiology* 186:627–635.

Levins R, Lewontin RC. 1980 Dialectics and reductionism in ecology. *Synthese* **43**, 47–78.

Levins R, Lewontin RC. 1994 Holism and reductionism in ecology. *Capitalism Nature Socialism*
5, 33–40.

Loken, E., and A. Gelman. 2017. Measurement error and the replication crisis. *Science* 355:584–
585.

MacKenzie DI, Nichols JD, Lachman GB, Droege S, Royle JA, Langtimm CA. 2002 Estimating
Site Occupancy Rates When Detection Probabilities Are Less Than One. *Ecology* **83**,
2248–2255.

Nathan R, Getz WM, Revilla E, *et al.* 2008. A movement ecology paradigm for unifying
organismal movement research. *Proc National Acad Sci* **105**: 19052–9.

Neyman, J., E. S. Pearson, and K. Pearson. 1933. IX. On the problem of the most efficient tests
of statistical hypotheses. Philosophical Transactions of the Royal Society of London.
Series A, Containing Papers of a Mathematical or Physical Character 231:289–337.

Nichols JD, Kendall WL, and Boomer GS. 2019. Accumulating evidence in ecology: Once is not
enough. *Ecol Evol* **4**: 220.

Nissen SB, Magidson T, Gross K, and Bergstrom CT. 2016. Publication bias and the
canonization of false facts. *Elife* **5**.

Open Science Collaboration. 2015. Estimating the reproducibility of psychological science.
Science 349:aac4716.

Platt JR. 1964. Strong inference. **146**.

Popper KR. 1959 *The logic of scientific discovery*. New York: Basic Books.

Quinn JF, Dunham AE. 1983 On Hypothesis Testing in Ecology and Evolution. *Am. Nat.* **122**,
602–617.

R Core Team. 2018 *R: A Language and Environment for Statistical Computing*. Vienna, Austria:
R Foundation for Statistical Computing. See <https://www.R-project.org/>.

Riotte-Lambert L, Matthiopoulos J. 2019 Environmental Predictability as a Cause and
Consequence of Animal Movement. *Trends Ecol. Evol.* (doi:10.1016/j.tree.2019.09.009)

Rothenberg, T. J. 1971. Identification in Parametric Models. *Econometrica: journal of the*
Econometric Society 39:577–591.

Sagan C. 1995. The fine art of baloney detection.

Sakiyama T, Gunji Y-P. 2016 Emergent weak home-range behaviour without spatial memory. **3**,
160214.

Servedio MR, Brandvain Y, Dhole S, *et al.* 2014. Not Just a Theory—The Utility of
Mathematical Models in Evolutionary Biology. *PLoS Biol* **12**: e1002017.

Sheil D, Burslem DFRP. 2013 Defining and defending Connell's intermediate disturbance
hypothesis: a response to Fox. *Trends Ecol. Evol.* **28**, 571–572.

Simberloff D. 1980 A succession of paradigms in ecology: essentialism to materialism and
probabilism. *Synthese* **43**, 3–39.

Stephens P, Buskirk S, and Rio C. 2007. Inference in ecology and evolution. *Trends Ecol Evol*
**22**: 192–7.

Tobias J. 2011. Hail the impossible: p- values, evidence, and likelihood. *Scand J Psychol* **52**:
113–25.

Vagle GL, McCain CM. 2020 Natural population variability may be masking the more-
individuals hypothesis. *Ecology* (doi:10.1002/ecy.3035)

Ward MP, Semel B, Jablonski C, Deutsch C, Giammaria V, Miller SB, M MB. 2011
Consequences of using Conspecific Attraction in Avian Conservation: A Case Study of
Endangered Colonial Waterbirds. *Waterbirds* **37**, 476–480.

Wasserstein RL and Lazar NA. 2016. The ASA's Statement on p-Values: Context, Process, and
Purpose. **70**: 129–33.

**FIGURE LEGENDS**

Figure 1. Conceptual flow chart of hypothesis vetting process. Researchers first specify
a set of candidate hypotheses to consider before writing them as formal models. Formal models
are checked for internal coherence and revised, if necessary. Sampling distributions of simulated
response variables are generated from each candidate hypothesis which can then be compared to
one another for evidence of degeneracy or noisiness. If no such problems exist, the researcher
proceeds with data-based inference. Alternatively, the researcher revises the set of candidate
hypotheses and begins the hypothesis vetting anew.

Figure 2. Testable Hypothesis ~~Heuristic~~ relationships between processes and observable
patterns that drive inferential outcomes. Boxes linked by arrows represent individual hypotheses
that can or can't be parsed based on observed patterns. Density plots show examples of sampling
distributions arising from each hypothesis. Processes Detectable from Patterns: for a hypothesis
to be testable, the response patterns must reliably, ~~and~~ quantifiably, and uniquely correspond to
the hypothesized mechanisms. Note how each process is uniquely linked to a distinct pattern
with little or no overlap between sampling distributions; Degenerate Relationship: multiple
mechanisms degenerating to an indistinguishable response pattern. Each unique process leads to
the same observation pattern and sampling distributions are almost completely overlapping.;
Noisy Relationship: a single mechanism does not reliably produce a concordant response pattern.
A single hypothesized process leads to a widely varying response pattern. High variance and/or
multi-modal sampling distribution makes estimation difficult or impossible.

Figure 3. Reproduced from (Fourcade et al., 2018) “Workflow used in analyses: 20
pseudo-predictors were created from the projection of paintings on the Western Palearctic
geographical space (examples: top: John Singer Sargent, Blonde Model, bottom: Zhang Daqian,
Spring dawns upon the colorful hills) and were used to compute species distribution models
(SDMs) after principal components analysis (PCA). A set of 20 true environmental variables
(climate and topography) was also used to compute SDMs for the same species. Both types of
models were evaluated using area under the receiver operating curve (AUC) and true skill
statistics (TSS). The SDMs presented at the bottom show the example of a species (*Candidula*
*unifasciata*, a land snail species) for which the SDM computed with pseudo-predictors led to
better evaluation metrics (here computed by randomly splitting occurrences into training and
testing datasets) than that computed with real environmental variables (suitability increases from
blue to red). AUC_p = AUC for model computed with paintings-derived pseudo-predictors;
AUC_e = AUC for model computed with real environmental variables; TSS_p = TSS for model
computed with paintings-derived pseudo-predictors; TSS_e = TSS for model computed with real
environmental variables”

Figure 4. Whisker plot of sampling distribution ranges for each parameterization of each
hypothesis used to detect noisy hypotheses. Wider whiskers indicate lower precision in
parameter estimation and potential evidence of a noisy relationship.

Figure 5. Heatmap of sampling distribution overlap. Panels clockwise from top-left:
p(HP|RAND) shows the proportion of habitat preference model simulations that overlapped the
range of null models for each parameterization; p(HP|CA) shows the proportion of habitat

preference model simulations that overlapped the range of conspecific attraction models for each
parameterization; p(CA|RAND) shows the proportion of conspecific attraction model
simulations that overlapped the range of null models for each parameterization; p(RAND|CA)
shows the proportion of null model simulations that overlapped the range of conspecific
attraction models for each parameterization; p(CA|HP) shows the proportion of conspecific
attraction model simulations that overlapped the range of habitat preference models for each
parameterization; p(RAND|HP) shows the proportion of null model simulations that overlapped
the range of habitat preference models for each parameterization.

Figure 6. Examples of spatial distributions of settled agents in two model iterations. Left:
A strong conspecific attraction parameterization. Note the very strong spatial clustering. Right:
Parameterized for habitat preference the model generates are more diffuse spatial pattern. While
both these models produced substantially overlapping sampling distributions, spatial metrics
could be used to parse hypotheses.

Figure 1-Figure 6.

Appendix B

Author response to reviewer comments on manuscript RSOS-200231.R1

We very much appreciate the reviewer's and editor's consideration of our revised manuscript. The comments we have received, including those discussed below, have helped improve this manuscript. Below we provide specific responses to each reviewer comment in **red text**; original referee comments are in *italics*.

Reviewer comments to Author:

Reviewer: 1

Comments to the Author(s)

Thank you for taking the time to address my comments. This is a great manuscript and I hope to see it published soon.

We appreciate the reviewer's comments and their earlier feedback on this manuscript.

Reviewer: 2

Comments to the Author(s)

Scientists embark on the journey of revealing true processes underlying observed patterns but there has been a considerable hurdle where inferences be an artifact of the practitioner's biases and decisions. This work continues the conversation about our collective responsibility to be conscious of and address possible shortcomings of our research. Particularly, we must test multiple hypotheses that may describe the mechanism underlying our observations. The paper provides a nice overview/review of this discussion over the decades and presents a contemporary solution to add to our toolkit as scientists, considering multiple hypotheses before data collection.

I reviewed an earlier version of this manuscript and I am very content with the revisions. The authors addressed all concerns and suggestions thoughtfully. I appreciate the effort the authors put in to these revisions.

I have no major comments. Their work will make a valuable contribution.

We very much appreciate this as well as earlier feedback this reviewer provided which has improved the manuscript.

Minutiae (None of these are critical to address)

L34: what is "true"? Perhaps just "discover possible relationships"

We decided to just remove the word “true” here since, as the reviewer pointed out, it presents epistemological problems. It now reads: “The ultimate goal of science is to learn about the relationships between observable patterns in the world around us...”

L87: Is the list of intentions brought up later? I think having two numbered lists might be a bit confusing especially so close together in your conclusions. Maybe labelling this list as Goal 1 and then the Steps starting on line 94 as Step 1 etc. if you refer to them.

Do the sections have to be numbered? I am trying to compare them to the other numbered lists. I know the numbering of headings may be based on journal format but it would be nice if only numbers were used intentionally and for repeatedly references lists.

We do not refer explicitly back to these goals in a way necessitating they be numbered. Therefore, we have removed the numbering from this list. Regarding headings, we see the mismatch between the numbered lists and the heading titles and agree it can lead to confusion. We do not believe the numbered sections are a journal requirement and have removed the numbering from the headings.

L127: Remove ‘Again’ since it is re-visited implying another visit unless this is the third visit

We do indeed mean to imply the (at least) third visit. This section is intended to outline the many times Chamberlin has been re-visited and to add ourselves to that conversation. As such, we would prefer to keep “again” in the heading title here.

L148: How does the researcher figure out how many hypotheses are plausible?

We suspect this comment refers to the section starting at line 184. It’s an astute point – one can never really know that they have the complete set of plausible hypotheses in hand. We imply this in a few places but agree that we could be a bit more explicit. To that end we have added the following sentence at line 206:

“In fact, there always remains the possibility of a plausible hypothesis a researcher has yet to consider.”

L246: “hypothesis; As the falsification” should be a period or not capitalized?

Changed the “a” to lowercase.

L340-341: Could the authors discuss these two questions like they do the third?

We added the following sentence after the first question:

“This will depend on e.g., expected effects sizes, intended uses of the research output, the scale at which the ecological process unfolds.”

The latter two questions were closely related and the explanation after them applies to both. We connected the two question into a single sentence with an em-dash to clarify.

L511: Betini et al 2017 is sometimes 2016 – please go through this and other references for consistency

Corrected and checked, as requested.